# A Deep-Learning Sequence-Based Method to Predict Protein Stability Changes Upon Genetic Variations

**DOI:** 10.3390/genes12060911

**Published:** 2021-06-12

**Authors:** Corrado Pancotti, Silvia Benevenuta, Valeria Repetto, Giovanni Birolo, Emidio Capriotti, Tiziana Sanavia, Piero Fariselli

**Affiliations:** 1Department of Medical Science, University of Turin, Via Santena 19, 10126 Torino, Italy; corrado.pancotti@unito.it (C.P.); silvia.benevenuta@unito.it (S.B.); valeria.repetto@edu.unito.it (V.R.); giovanni.birolo@unito.it (G.B.); piero.fariselli@unito.it (P.F.); 2Department of Pharmacy and Biotechnology (FaBiT), University of Bologna, Via Francesco Selmi 3, 40126 Bologna, Italy; emidio.capriotti@unibo.it

**Keywords:** deep learning, protein stability, free energy changes, antisymmetry, ACDC, sequence

## Abstract

Several studies have linked disruptions of protein stability and its normal functions to disease. Therefore, during the last few decades, many tools have been developed to predict the free energy changes upon protein residue variations. Most of these methods require both sequence and structure information to obtain reliable predictions. However, the lower number of protein structures available with respect to their sequences, due to experimental issues, drastically limits the application of these tools. In addition, current methodologies ignore the antisymmetric property characterizing the thermodynamics of the protein stability: a variation from wild-type to a mutated form of the protein structure ($X_{W}\to X_{M}$) and its reverse process ($X_{M}\to X_{W}$) must have opposite values of the free energy difference ($\Delta \Delta G_{WM}=-\Delta \Delta G_{MW}$). Here we propose ACDC-NN-Seq, a deep neural network system that exploits the sequence information and is able to incorporate into its architecture the antisymmetry property. To our knowledge, this is the first convolutional neural network to predict protein stability changes relying solely on the protein sequence. We show that ACDC-NN-Seq compares favorably with the existing sequence-based methods.

## 1. Introduction

Predicting protein stability changes upon genetic variations is still an open challenge. It is essential to understand the impact of the alterations in the amino acid sequence, mainly due to non-synonymous (or missense) DNA variations leading to the disruption or the enhancement of the protein activity, on human health and disease [1,2,3,4]. In particular, protein stability perturbations have already been associated to pathogenic missense variants [5] and they were shown to contribute to the loss of function in haploinsufficient genes [6].

The protein stability changes upon variations of the amino acid sequence is usually expressed as the Gibbs free energy of unfolding (ΔΔG), which is defined as the difference between the energy of the mutated structure of the protein and its wild-type form ($\Delta \Delta G=\Delta G_{M}-\Delta G_{W}$). Thermodynamics imposes an antisymmetry relationship on ΔΔG that can be summarized as follows: given the wild-type (W) and mutated (M) protein structures, differing by one residue in position X, the quantity $\Delta \Delta G_{WM}\left(=\Delta G_{W}-\Delta G_{M}\right)$ represents the change in the protein stability caused by the amino acid substitution $X_{W}\to X_{M}$. Similarly, given the symmetry between the two molecular systems M and W, for the reverse variation $X_{M}\to X_{W}$ the corresponding change in Gibbs free energy has the opposite sign:(1)$$\Delta \Delta G_{WM}=-\Delta \Delta G_{MW}.$$

Since experimental measurement of ΔΔG is a time-consuming and complex task, during the last decades several computational tools have been developed to predict ΔΔG values. Some methods are structure-based, requiring the knowledge of the protein tertiary structure [7,8,9,10,11,12], others are sequence-based, either relying only on protein sequences [13,14,15] or optionally taking advantage of the protein structure when available [16,17]. However, most of these methods violate the antisymmetry property and suffer from high biases in predicting reverse variations [10,18,19,20,21,22,23]. To address this problem we recently introduced ACDC-NN, a novel structure-based method that satisfies the physical property of antisymmetry, while reaching comparable performance to the state-of-the-art methods [24]. However, the experimental structure determination and characterization of protein thermodynamical features are still limited [18], while a dramatic increase in protein sequence databases has occurred as genomic and metagenomic sequencing efforts have expanded in the last years. The latest release of UniProtKB/TrEMBL protein database contains 214,406,399 sequence entries in all, including 175,817 human proteins, while the Protein Data Bank contains 177,655 entries, 52,485 of them human. Hence, computational approaches able to predict the impact of genetic variations on the protein stability using only sequence information are needed. To this aim, we created ACDC-NN-Seq, a sequence-based version of ACDC-NN that, like its predecessor, achieves accurate predictions satisfying the antisymmetry property, without the need for tertiary protein structures. Here we show that ACDC-NN-Seq compares well with both sequence-based and structure-based methods. We tested the antisymmetry of the predictions on an unbiased dataset and the accuracy on three clinically relevant proteins, avoiding overfitting by filtering out sequence similarities greater than 25%.

## 2. Materials and Methods

### 2.1. Datasets and Cross-Validation

Since the application of a deep learning technique requires a large amount of data to achieve the best performance, we pre-trained ACDC-NN-Seq using the predictions of another method, DDGun3D, which has shown to achieve antisymmetry with good performance. DDGun3D was not trained on experimental data, but it is based on evolutionary information and statistical potentials [14]. The pre-training phase was performed on an artificial set of variants created from the Ivankov dataset [21], that we named IvankovDDGun. The ΔΔG values for this dataset were calculated using DDGun3D and the obtained scores were learnt by ACDC-NN-Seq.

For training/testing the network we considered the observed experimental ΔΔG variants reported by some of the most widely used datasets extracted from Protherm [25] database, already cleaned for redundancies and inaccuracies that are known to affect this database, which are: S2648 [26] and Varibench [27]. The ACDC-NN-Seq performance was tested on the following datsets: Ssym [26], p53 [28], myoglobin [29] and frataxin mutants from the CAGI5 challenge [30]. The datasets are summarized in Table 1 and a complete description can be found in our previous study [24].

Since these datasets often contain proteins with a high degree of sequence similarity between them, we avoided the related overfitting issues by filtering out similarities above 25%, as done in ACDC-NN; we generated cross-validation folds with blastclust algorithm [31] to create clusters of protein with sequence identity lower than 25% (command blasclust -i infile.fasta -o out.custers -p T -L 0.5 -b F -S 25).

### 2.2. Sequence Profiles

ACDC-NN-Seq uses the evolutionary information derived from multiple sequence alignments. Through the alignment it is possible to obtain a profile, i.e., an N × 20 matrix, where each entry Prof(i,j) corresponds to the frequency of the *j*-th residue in the *i*-th sequence position. N represents the protein length and 20 are the different aminoacids residues. Multiple sequence alignments were computed using the Uniprot database (release 2016) by the *hhblits* tool [32] with default parameters.

### 2.3. ACDC-NN-Seq Architecture

The ACDC-NN-Seq architecture is the sequence-based counterpart of the 3D version, already published [24]. Here, we report only the relevant points, leaving out some mathematical details.

ACDC-NN-Seq is an Antisymmetric Convolutional Differential Concatenated Neural Network (ACDC-NN) that takes as inputs both direct and reverse variations, processes them by convolution operations and then uses the extracted features as input for two siamese neural networks [33,34]. Specifically, each of the two ACDC-NN-Seq inputs consists of 160 elements to code variation and sequence evolutionary information:**Variation (V)**: 20 features (one for each amino acid) coding for the variation by setting all the entries to 0 with the exception of the wild-type and the variant residue positions set to −1 and 1, respectively. This input corresponds to a one-dimensional matrix $V\in R^{20\times 1}$;**Sequence (S or 1D-input)**: 140 features representing protein profile information of the variation neighbourhood. Considering *i* as the variant position in the sequence, we used a window of 3 residues, i.e., [i−3;i+3], so to obtain 20×7 elements, with the profile information of these 7 positions. This input then corresponds to a sequence of 7 vectors taken from the protein profile.

A 2D Convolutional layer was applied on the *S* matrix using a kernel equal to (1,20) and stride (1,1) (reported as Keras-style parameters [35]) and generating a 20×20 filter matrix $K_{S}$. After the convolution, a dot product was performed between the variant vector V and the resulting 2D convolution matrix, obtaining 7 processed features, computed for both the direct variation and its reverse. These features were then concatenated with the variant vector *V* and used as input to a Differential Siamese Networks [33,34]. Finally, their outputs were combined in two Lambda layers of “difference/2” and average. To incorporate the antisymmetric property into the network structure, we designed a specific loss function that minimizes both the absolute value of the average output and the distance between the difference output and the true ΔΔG values. The ACDC-NN-Seq architecture is displayed in Figure 1 and Figure 2.

The choice of an input window of 7 residues, three for each side of the position of interest, was made to learn the DDGun3D mapping (in the pre-training phase) since we adopted the same DDGun3D sequence neighborhood. Although we experimented with some other input window sizes, there was not much difference up to 11 residues, where we found a performance decrease.

### 2.4. Pre-Training Phase

Due to the lack of experimental data to train a network from scratch, we first pre-trained the model on an artificial dataset and then performed transfer learning on real datasets. The pre-training phase was performed on the unlabeled artificial dataset IvankovDDGun, which includes all the possible direct and reverse variations in every sequence position. We chose to perform the predictions on the IvankovDDGun dataset with a 3D method to internally encode 3D information from the sequence; DDGun3D was selected among the other predictors for its near-perfect antisymmetry. The training, validation and test sets used by DDGun3D include 400,000, 100,000 and 100,000 protein variants, respectively. The Differential Siamese Network consists of two hidden layers with 128 and 64 units (complete description in Table 2).

### 2.5. Transfer Learning on Experimental Data

After the pre-training phase, we applied transfer learning using S2648 [26] and Varibench [27] datasets of experimental ΔΔG values, splitting the data and removing the sequence similarity among training, validation and test sets. Specifically, the weights of the Convolutional layer were fixed while the Differential Siamese network part was re-trained selecting the best parameters. To increase the size of the training set, the unlabeled Ivankov2000 dataset with DDGun3D predictions was also considered. The complete description of the optimal sets of parameters is shown in Table 2.

### 2.6. Performance Evaluation

Pearson correlation (indicated by *r*) and root mean square error (RMSE) were estimated between the predicted and observed ΔΔG values to evaluate the performance of the methods. Two scoring indices were adopted to assess the antisymmetric property of ΔΔG predictors: $r_{d-i}$ and 〈δ〉. $r_{d-i}$ is the Pearson correlation coefficient between the direct and the corresponding reverse variations:(2)$$\begin{array}{c} r_{d-i}=\frac{Cov(\Delta \Delta G^{dir},\Delta \Delta G^{inv})}{\sigma_{dir}\sigma_{inv}} \end{array}$$
where Cov is the covariance and σ is the standard deviation. 〈δ〉 is the average bias quantifying the prediction shift:(3)$$\langle \delta \rangle =\frac{\sum_{i=1}^{N}\left(\Delta \Delta G_{i}^{dir}+\Delta \Delta G_{i}^{inv}\right)}{2N}.$$
A perfectly antisymmetric method should have $r_{d-i}$ equal to −1 and 〈δ〉 equal to 0.

## 3. Results

### 3.1. Learning 3D Properties on Artificial Data

A proper training of a neural network requires a huge amount of experimental ΔΔG values that are not currently available; we addressed this problem by performing a pre-training phase on the artificial dataset IvankovDDGun and then applying a transfer learning on the experimental datasets S2648 and Varibench, as described in the Materials and Methods section.

In Table 3, we show the results on the IvankovDDGun test set. It is worth noticing that the DDGun3D values were computed using the protein structures, while the ACDC-NN-Seq predictions are only based on sequence information. Thus, this approach obtained a sequence-based method capable of internally encoding the 3D statistical potentials that maintain the antisymmetric property.

### 3.2. Prediction of the Experimental ΔΔG Values

After training ACDC-NN-Seq on the IvankovDDGun set, we fine-tuned the network by retraining the last layers on the experimentally-derived ΔΔG values from S2648 and Varibench through 10-fold cross-validation. In Figure 3 we showed the experimental ΔΔG values versus the ACDC-NN-Seq predicted ones on Varibench and S2648 datasets both combined and alone, and for both direct and reverse variations. These results were obtained in cross-validation as explained in Benevenuta et al. [24]. ADCD-NN-Seq achieved both consistent performance with the state-of-the-art methods (measured in terms of *r* and RMSE) and perfect antisymmetry ($r_{d-i}=-1$ and 〈δ〉=0.0).

We also compared the predictions of both ACDC-NN-Seq and DDGun with their corresponding stucture-based versions, i.e., ACDC-NN and DDGun3D. Figure 4 reports the comparison performance (in cross-validation for ACDC-NN and ACDC-NN-Seq) on the Ssym dataset [10], which was specifically built to assess the antisymmetry and it contains ΔΔG experimental values for direct and reverse variants. The performance of ACDC-NN-Seq is balanced and close to those obtained using the protein structures. This makes ADCD-NN-Seq ideal for genome variant analyses.

In order to evaluate the effect of the neural network design, we compared ACDC-NN-Seq with a feed-forward neural network (FFNN) trained and optimized in the same conditions. The structure of the optimized FFNN consists of an input layer of 140 input neurons (window of 7 residues coded with 20-element vector profiles), a sequence of hidden layers consisting of (128,64,32,16) neurons, and an output neuron coding for the ΔΔG value. Thus the main difference is due to the anty-symmetric construction of ACDC-NN-Seq (FFNN Figure 4). FFNN performance is quite good, and the neural networks learned most of the antisymmetry from the data provided (direct and reverse variations). However, ACDC-NN-Seq outperforms FFNN both in the prediction task and antisymmetry reconstructions.

Regarding the results presented in Figure 3 and Figure 4, it is worth noticing that the maximum achievable Pearson’s correlation is not necessarily equal to 1, as usually thought. It may be far lower depending on the experimental uncertainty and the ΔΔG distributions [36,37]. In particular, when considering the different experiments on the same variants included in the Protherm database or in manually-cleaned datasets, the expected Pearson upper bound is in the range of 0.70–0.85 [36]. Significantly higher Pearson correlations can be obtained in small sets or might be indicative of overfitting issues [36].

### 3.3. Comparison with Other Sequence-Based Machine-Learning Methods

As mentioned above, few available methods can predict the effect of the variants on the protein stability starting from sequence only. We therefore compared ACDC-NN-Seq on three datasets with the following sequence-based methods: DDGun [14], INPS [13], I-Mutant2.0 [16], MUpro [17] and the recent SAAFEC-SEQ [15].

The obtained results are reported in Table 4; ACDC-NN-Seq predicts equally well both direct and reverse variants with nearly perfect antisymmetry (−0.99). ACDC-NN-Seq performance is higher than the one obtained by INPS, which is the only machine-learning method proven to be antisymmetric in past tests [22,38].

I-Mutant2.0 and MUpro do not respect the antisymmetry property since this issue was not properly addressed or known at the time the two models were created. However, it must be noted that both I-Mutant2.0 and MUpro do not use evolutionary information making them extremely fast predictors, as compared to ACDC-NN-Seq, which requires a multiple sequence alignment.

Another significant point is that, looking at all the methods reported in Table 4, Table 5 and Table 6, only Inps-NoSeqId and all the versions of ACDN-NN were trained in cross-validation removing the sequence identity (i.e., sequence similarity <25%).

### 3.4. Frataxin CAGI 5 Challenge

The Critical Assessment of Genome Interpretation (CAGI) is a community experiment aimed at fairly assessing the computational methods for genome interpretation [30]. In CAGI 5, data providers measured unfolding free energy of a set of variants with far-UV circular dichroism and intrinsic fluorescence spectra on Frataxin (FXN), a highly conserved protein fundamental for the cellular iron homeostasis in both prokaryotes and eukaryotes. These measurements were used to calculate the change in unfolding free energy between the variant and wild-type proteins at zero denaturant concentrations (ΔΔG). In addition, the experimental dataset [39], including eight amino acid substitutions, was used to evaluate the performance of the web-only tools, based on protein structure information, for predicting the value of the associated ΔΔG [40]. Here we compare the available machine-learning sequence-based predictors on the dataset (Table 7), showing the consistency of the prediction performance of ACDC-NN-Seq.

## 4. Discussion

Few sequence-based methods are currently available to predict the ΔΔG and only some of them satisfy the principle of antisymmetry imposed by thermodynamics. Given the huge amount of sequencing data currently available and their possible applications, we have proposed a reliable sequence-based predictor that could be applied to a wide range of biological problems, even when the crystallized protein structures are not available. Moreover, ACDC-NN-Seq addresses the antisymmetry physical property by exploiting a specifically-designed loss function which minimizes the absolute difference between the direct and reverse predictions [24]. We showed that ACDC-NN-Seq compares well with with the other state-of-the-art sequence-based machine-learning predictors and with some of the structure-based ones, as shown in Table 4, Table 5, Table 6 and Table 7. To avoid overfitting issues due to sequence similarity, the correct procedure to assess the performance of a method on any dataset should use variants in proteins which are not similar to those used for the training [18]. Specifically, this means to remove proteins with a sequence identity >25% between the training and testing data. In this study, we followed this procedure and we divided all the datasets used into 10 non-similar subsets during cross-validation. In conclusion, ACDC-NN-Seq is a sequence-based tool able to show comparable or better performance with respect to the state-of-the-art methods while preserving perfect thermodynamic antisymmetry. In future studies, this method can be extended to predict the ΔΔGs of multiple site variations.

## Figures and Tables

**Figure 1 genes-12-00911-f001:**
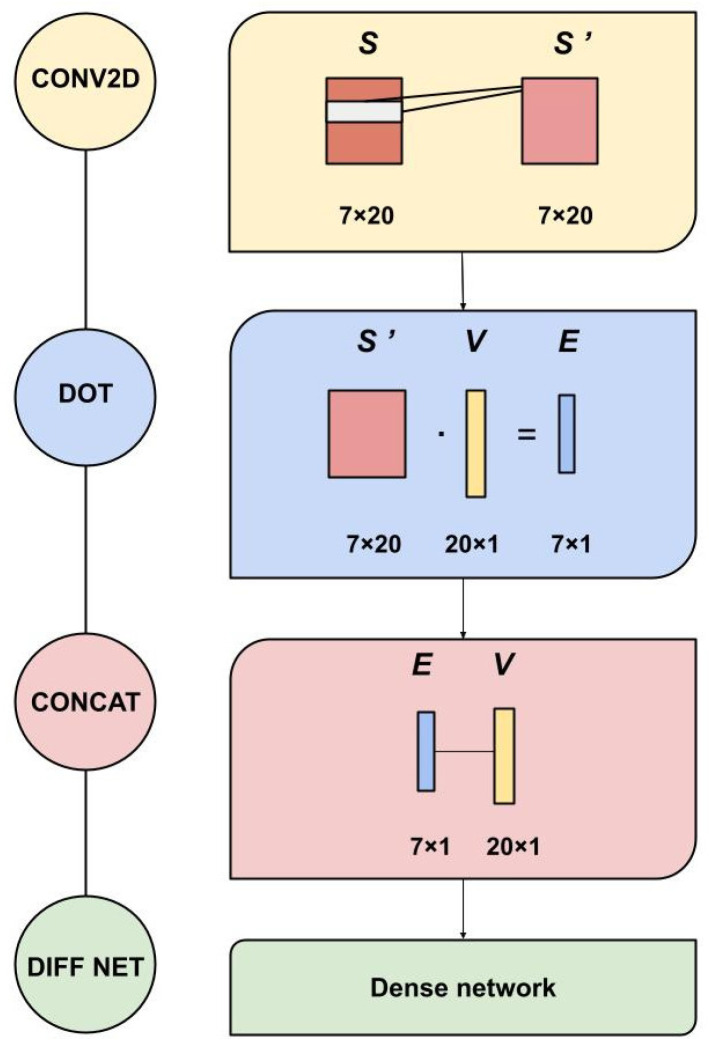
Constituent modules of ACDC-NN-Seq. CONV2D: a 2D-convolution operation applied to S (1D inputs) with 20 filters with kernel (1,20) and stride (1,1); DOT: dot product is applied to S’ processed information with the variation encoding vector V; CONCAT: all the 27 features are concatenated and used as input in a Dense Network (DIFF NET).

**Figure 2 genes-12-00911-f002:**
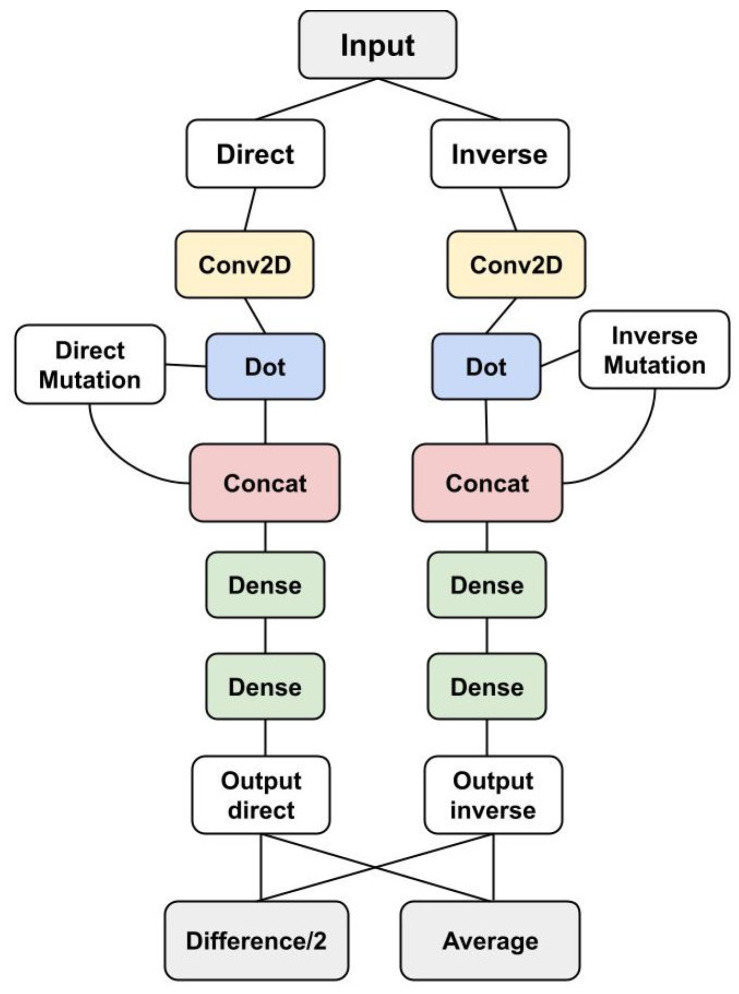
Complete ACDC-NN-Seq architecture. The module displayed in Figure 1 is used for both direct and reverse variations. Given a variation, we provide to the network its coding to the left, and the coding to the reverse variation to the right. A final layer takes the average and the difference between the two outputs. The difference computes $\left(\Delta \Delta G_{direct}-\Delta \Delta G_{inverse}\right)$/2, which in case of perfect antisymmetry is exactly equal to ΔΔG. The average computes $\left(\Delta \Delta G_{direct}+\Delta \Delta G_{inverse}\right)$/2, which in case of perfect antisymmetry is equal to 0. The ACDC-NN-Seq outputs are estimations of the target ΔΔGs learned during the training phase. The two Siamese networks have shared weights, both in the convolutional and the dense parts of the network.

**Figure 3 genes-12-00911-f003:**
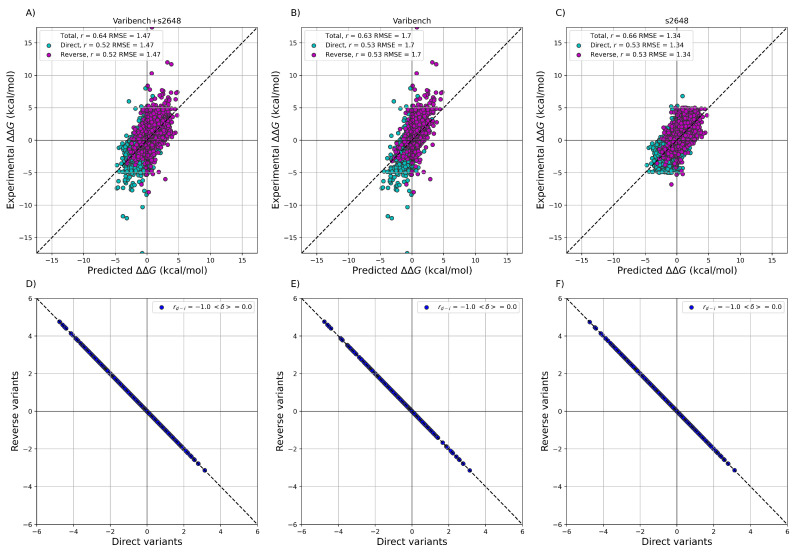
Performance of ACDC-NN-Seq on predicting ΔΔG for the direct and reverse variations on: (**A**) Varibench and S2648 (r=0.52, RMSE=1.47 kcal/mol); (**B**) Varibench alone (r=0.53, RMSE=1.7 kcal/mol); (**C**) S2648 alone (r=0.53, RMSE=1.34 kcal/mol). Direct versus reverse ΔΔG values of (**D**) Varibench and S2648 variations, (**E**) Varibench variations alone, (**F**) S2648 variations alone, predicted by ACDC-NN-Seq, with a Pearson correlation of $r_{d-i}=-1.0$ and 〈δ〉=0.00 kcal/mol for all three datasets. All the predictions reported in this figure were obtained through a 10-fold cross-validation with sequence identity <25% among all folds.

**Figure 4 genes-12-00911-f004:**
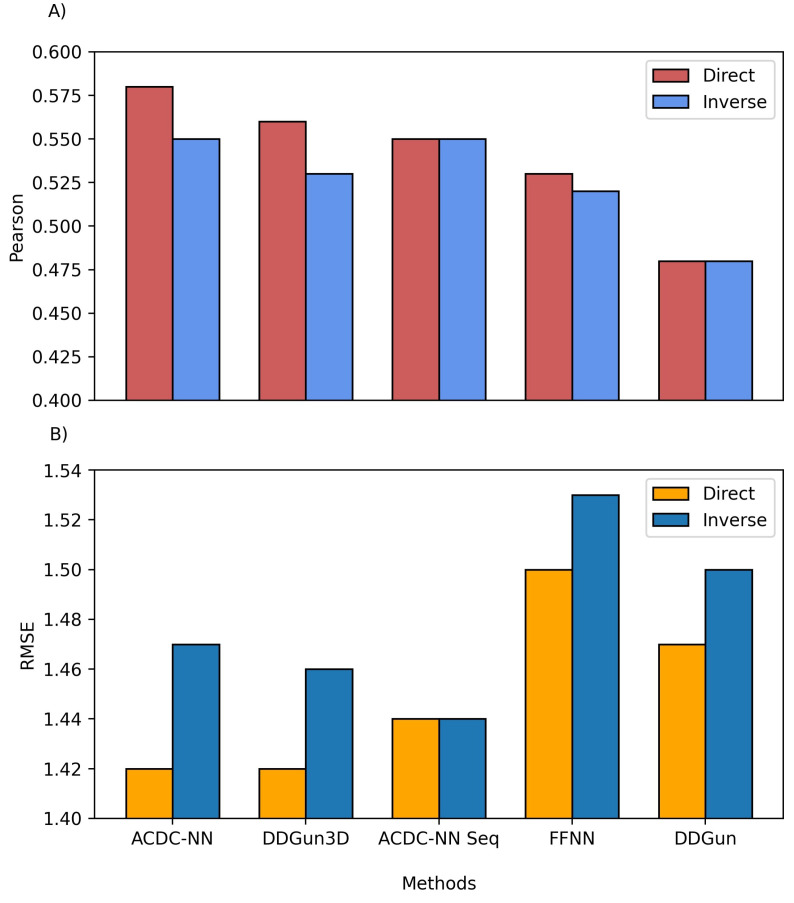
Comparison between the structure and sequence-based versions of ACDC-NN, DDGun and a Feed-Forward Neural Network (FFNN) on the Ssym dataset. (**A**) Pearson correlation coefficient (*r*), where higher is better and (**B**) root mean squared error (RMSE), where lower is better.

**Table 1 genes-12-00911-t001:** **Brief description of the datasets used.** The table reports the number of variants available on each dataset, their usage in this study and whether or not the experimental ΔΔG values are available. DDGun3D was used for the estimation of ΔΔGs when they are not available.

Dataset	Number of Variants	Usage	Experimental ΔΔG
IvankovDDGun	600,000	Pre-Training	No
Ivankov2000	2000	Transfer Learning	No
S2648	2648	Transfer Learning	Yes
Varibench	1420	Transfer Learning	Yes
Ssym	684	Test	Yes
Myoglobin	134	Test	Yes
p53	42	Test	Yes
Frataxin-CAGI	8	Test	Yes

**Table 2 genes-12-00911-t002:** The final architectures of the Differential Siamese Network before and after transfer learning. The optimal parameters were selected on a validation set without intersections with the training set.

NN Parameters	Before Transfer-Learning	After Transfer-Learning
Hidden units	12,864	12,864
Dropout	0.05	0.35
Epochs	45	30
Batch-size	500	150
Optimizer	Adam	Adam
Loss	logcosh + abs	logcosh + abs

**Table 3 genes-12-00911-t003:** **Results on the IvankovDDGun test set:** The performance of ACDC-NN-Seq in learning DDGun3D was measured in terms of Pearson correlation coefficient (r) and root mean square error (RMSE). The antisymmetry property was assessed in terms of Pearson correlation coefficient ($r_{d-i}$) and the bias (〈δ〉) between the predicted values. RMSE and 〈δ〉 are expressed in kcal/mol. IvankovDDGun (Test) is the test set extracted from the IvankovDDGun artificial dataset.

Dataset	Pearson/RMSE		Antisymmetry	
	Direct	Reverse	$r_{d-i}$	〈δ〉
IvankovDDGun (Test)	0.97/0.06	0.97/0.06	−1.0	0.0

**Table 4 genes-12-00911-t004:** **Results on Ssym:** The performance on both direct and reverse variants was measured in terms of Pearson correlation coefficient (r) and root mean square error (RMSE). The antisymmetry was assessed using the correlation coefficient $r_{d-i}$ (Equation 2) and the bias 〈δ〉 (Equation 3). RMSE and 〈δ〉 are expressed in kcal/mol. The results of INPS were taken from Montanucci et al. [14] and Fariselli et al. [13]; the results of SAAFEC-SEQ and I-mutant2.0 were obtained using their stand-alone code, those of MUpro were obtained using the webserver available. Only Inps-NoSeqId and ACDN-NN-Seq were trained in cross-validation addressing the sequence identity issue (sequence similarity <25%).

Method	Pearson/RMSE		Antisymmetry	
	Direct	Reverse	$r_{d-i}$	〈δ〉
ACDC-NN-Seq	0.55/1.44	0.55/1.44	−0.99	−0.01
INPS-NoSeqId [22]	0.48/1.42	0.47/1.45	−0.99	−0.06
INPS [13]	0.51/1.42	0.50/1.44	−0.99	−0.04
SAAFEC-SEQ [15]	0.71/1.09	−0.39/2.71	0.58	−1.84
I-Mutant2.0 [16]	0.7/1.12	0.05/2.54	−0.17	−1.01
MUpro [17]	0.79/0.94	0.07/2.51	−0.02	−0.97

**Table 5 genes-12-00911-t005:** **Results on myoglobin:** Comparison on myoglobin. The INPS, SAAFEC-SEQ and I-mutant2.0 results were obtained using their stand alone code, those of MUpro were obtained using the webserver available.

Method	Pearson/RMSE		Antisymmetry	
	Direct	Reverse	$r_{d-i}$	〈δ〉
ACDC-NN-Seq	0.56/0.97	0.56/0.97	−1.00	0.00
INPS	0.60/0.99	0.61/0.98	−1.00	0.01
SAAFEC-SEQ	0.63/0.89	0.30/1.63	−0.21	−1.50
I-Mutant2.0	0.56/1.12	0.39/1.71	−0.45	−0.88
MUpro	0.51/0.99	0.35/1.75	−0.17	−0.79

**Table 6 genes-12-00911-t006:** **Results on p53:** Comparison on p53. The INPS, SAAFEC-SEQ and I-mutant2.0 results were obtained using their stand alone code, those of MUpro were obtained using the webserver available.

Method	Pearson/RMSE		Antisymmetry	
	Direct	Reverse	$r_{d-i}$	〈δ〉
ACDC-NN-Seq	0.62/1.62	0.62/1.62	−1.00	0.00
INPS	0.72/1.49	0.70/1.54	−0.99	−0.01
SAAFEC-SEQ	0.52/1.64	−0.18/2.97	0.06	−1.79
I-Mutant2.0	0.35/1.75	0.22/2.81	−0.24	−1.02
MUpro	0.23/1.78	0.04/2.87	0.12	−0.98

**Table 7 genes-12-00911-t007:** **Results on Frataxin Challenge in CAGI5** [30]. The INPS, SAAFEC-SEQ and I-mutant2.0 results were obtained using their stand alone code, those of MUpro were obtained using the webserver available.

Method	Pearson/RMSE		Antisymmetry	
	Direct	Reverse	$r_{d-i}$	〈δ〉
ACDC-NN-Seq	0.88/2.83	0.88/2.83	−1.00	0.00
INPS	0.65/3.29	0.57/3.38	−0.99	−0.01
SAAFEC-SEQ	0.67/3.3	0.1/4.85	0.2	−1.94
I-Mutant2.0	0.84/2.82	0.53/5.08	−0.74	−1.22
MUpro	0.33/3.6	0.13/4.97	−0.23	−0.45

## Data Availability

The data presented in this study and the code to reproduce the results are freely available in https://github.com/compbiomed-unito/acdc-nn (accessed on 11 June 2021).

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
