# Peer review of "A Deep-Learning Sequence-Based Method to Predict Protein Stability Changes Upon Genetic Variations"

_genes, 2021, doi:10.3390/genes12060911_

Round 1
Reviewer 1 Report
Authors present a deep-learning based approach for prediction of protein stability changes relying on protein sequences. The approach is quite novel and the study is quite sound.
Minor comments:
- Authors have not compared their approach against some newer sequence based approaches like SAAFEC-SEQ.
- It would be very valuable to see the performance of the new approach on the independent CAGI dataset.
- Comparison of the provided DL-architecture with other possible DL-architectures and comparison against traditional ML approaches on their own framework would make this study more interesting. In other words, there is no rational provided for choosing the particular type of DL-architecture for the proposed work.
Author Response
Reviewer 1.
Authors present a deep-learning based approach for prediction of protein stability changes relying on protein sequences. The approach is quite novel and the study is quite sound.
A: We thank the reviewer for her/his comments.
Minor comments:
-
Authors have not compared their approach against some newer sequence based approaches like SAAFEC-SEQ.
A: We thank the reviewer for having pointed out the new SAAFEC-SEQ tool. SAAFEC-SEQ has been included as a comparison in the manuscript, now in Tables 4-7. From the obtained values, it is clear that SAAFEC-SEQ is not compliant with the thermodynamics DDG requirements and it is biased towards destabilizing variants.
-
It would be very valuable to see the performance of the new approach on the independent CAGI dataset.
A: We thank the reviewer for the suggestion. We included the comparison of the sequence-based methods on the Frataxin CAGI dataset, Table 7 of the revised version. As the reviewer can see the results are very encouraging.
-
Comparison of the provided DL-architecture with other possible DL-architectures and comparison against traditional ML approaches on their own framework would make this study more interesting. In other words, there is no rational provided for choosing the particular type of DL-architecture for the proposed work.
A: We thank the reviewer for the suggestion. In the amended version we included a feed-forward neural network optimized in the same conditions as ACDC-NN-Seq. The network performs reasonably well but not at the same level, indicating that the anti-symmetry by construction can help the model to find a better generalization. In the revised version of the manuscript, we reported the results of this network (FFNN) in Figure 4.
Reviewer 2 Report
First deep-learning approach to predict protein stability changes uniquely relying on the protein sequence. But I believe that INPS is also a machine learning method, No?
The input is transformed into a matrix (7x20) which is a profile with a window of 3. I notice that in some previous papers of the group, they used a window of 2. It would be nice it they comment the influence of the size of the window on their results.
It is unclear if there is such a matrix for each position i of the N amino acids of the sequence or not. In other words, it is difficult to figure out where is the dimension of the sequence. This should be specified.
At least some references should be given for non specialists, concerning: Keras method, Differential Siamese Network
In the flow chart of Figure 2, it is unclear why two steps of dense are performed. Besides, I understand from this legend, that the output is a DDG calculation. But, until the previous step, one deals with sequences and some variable related to the probability of mutation. How is calculated the free energy is not specified. I guess these data are extracted from the training set of experimental value, but it should be commented.
I am not able to validate the parameters presented in Table 2. Nevertheless, it is mentioned that the training set is increased : how is it increased?). Actually, after a second reading, the sentence that makes problem, to my eyes, is the following, page 5:
“To increase the size of the training set, the unlabeled Ivankov2000 dataset with DDGun3D predictions was also considered “
Ivankov2000 is presented as a dataset with no DDG experimental value; so, what is the interest for a training set? Maybe I missed the relation between Ivankov2000 and DDGun3D (If so, I guess other readers will also miss it). Later it is specified that Ivankov2000 is derived from predictions of DDGun3D, so is there a putative bias between learning and test datasets?
Please, consider rewriting it in a more intelligible manner. The expression “transfer learning” (and also the meaning of unlabeled) should be developed for better understanding.
I also have the question about the datasets used: many papers rely on the Protherm database, which is a popular benchmark. Why the authors do not mention use it in this study? I do understand that they are comparing to predictors, themselves using Protherm as a gold standard, but a few comments could be added.
Page 8, the claim that a Pearson coefficient is not necessarily 1 for testifying of the quality of a predictor should be more developed. The argument is some sort of self-fulfilling prophecy.
Page 8, I understand that ACDC-NN-Seq is rather low in terms of computer time. It could be useful to give some values compared to the cited programs (at least for the stand-alone versions).
I guess there is an error in the legend of Table 4 because ACDC-NN is cited twice.
Honestly, the performance of ACDC-NN-Seq is of the same quality of INPS and the expression outperform, mentioned several times, is slightly exaggerated.
Author Response
Reviewer 2.
First deep-learning approach to predict protein stability changes uniquely relying on the protein sequence. But I believe that INPS is also a machine learning method, No?
A: The reviewer is correct. We intended a “convolutional neural network”. We changed the word accordingly in the abstract.
The input is transformed into a matrix (7x20) which is a profile with a window of 3. I notice that in some previous papers of the group, they used a window of 2. It would be nice it they comment the influence of the size of the window on their results.
A: We thank the reviewer for the question. Since during the pre-training we wanted to learn the DDGun3D mapping, we adopted the same local sequence context [mutation position +/-3). During the ACDC-NN tuning, we experimented with some other input windows. However, there was not much difference up to +/-5, and then there was a decrease in the performance. We added this information in the amended version of the manuscript (section 1.3).
It is unclear if there is such a matrix for each position i of the N amino acids of the sequence or not. In other words, it is difficult to figure out where is the dimension of the sequence. This should be specified.
A: Probably we were not clear enough in the manuscript. We thank the reviewer for having noticed it. In the revised version we explained better that we have considered only a window around the residue that undergoes a variation.
At least some references should be given for non specialists, concerning: Keras method, Differential Siamese Network
A: In the revised version of the manuscript we added the references.
In the flow chart of Figure 2, it is unclear why two steps of dense are performed. Besides, I understand from this legend, that the output is a DDG calculation. But, until the previous step, one deals with sequences and some variable related to the probability of mutation. How is calculated the free energy is not specified. I guess these data are extracted from the training set of experimental value, but it should be commented.
A: We changed the text to improve the readability. The network was trained to predict the DDG values, which were provided as target input during the training phase. No potential was explicitly computed internally to the network. However, they could be learned by the NN.
I am not able to validate the parameters presented in Table 2. Nevertheless, it is mentioned that the training set is increased : how is it increased?). Actually, after a second reading, the sentence that makes problem, to my eyes, is the following, page 5: “To increase the size of the training set, the unlabeled Ivankov2000 dataset with DDGun3D predictions was also considered “. Ivankov2000 is presented as a dataset with no DDG experimental value; so, what is the interest for a training set? Maybe I missed the relation between Ivankov2000 and DDGun3D (If so, I guess other readers will also miss it). Later it is specified that Ivankov2000 is derived from predictions of DDGun3D, so is there a putative bias between learning and test datasets?
Please, consider rewriting it in a more intelligible manner. The expression “transfer learning” (and also the meaning of unlabeled) should be developed for better understanding.
A: We maintained the training and testing without sequence identity in all cases. Therefore, in the revised version of the manuscript, we explain the pretraining procedure. We hope now it will be more understandable.
I also have the question about the datasets used: many papers rely on the Protherm database, which is a popular benchmark. Why the authors do not mention use it in this study? I do understand that they are comparing to predictors, themselves using Protherm as a gold standard, but a few comments could be added.
A: We thank the reviewer for having pointed out the missing information. We made a mistake, not recognizing the origin of most of the data used. The two larger datasets, S2648 and Varibench, are actually subsets of Protherm. They were manually cleaned from the original database since there were some inconsistencies and mistakes in Protherm. In the amended version of the manuscript we explained it.
Page 8, the claim that a Pearson coefficient is not necessarily 1 for testifying of the quality of a predictor should be more developed. The argument is some sort of self-fulfilling prophecy.
A:The rationale behind the claim has been explained in two published papers [ref montanucci et al, 2018 and Benevenuta and Fariselli 2019]. In particular, using Protherm with multiple experiments on the same variants, the computed upper bound is close to the simulated values and is in the range 0.70-0.85, for the Pearson coefficient. If more data with narrowed experimental uncertainty will be available in the future, the upper bound can be shifted towards one. The upper bound is also useful to highlight overfitting in papers claiming too high Pearson correlation values. We added this information in the revised version of the text.
Page 8, I understand that ACDC-NN-Seq is rather low in terms of computer time. It could be useful to give some values compared to the cited programs (at least for the stand-alone versions).
A: The limiting step is due to the multiple sequence alignment, and it is shared with all the tools that exploit evolutionary information. It depends on the sequence and the database, and it can be from a few seconds to hours. It is not predictable in advance.
I guess there is an error in the legend of Table 4 because ACDC-NN is cited twice.
A: We thank the reviewer for having noticed it. Yes, there was a mistake in the caption. We corrected it in the revised version.
Honestly, the performance of ACDC-NN-Seq is of the same quality of INPS and the expression outperform, mentioned several times, is slightly exaggerated.
A: The reviewer is right and we smoothed the sentence accordingly.